# Homeostasis in *C. elegans* sleep is characterized by two behaviorally and genetically distinct mechanisms

**Stanislav Nagy[1†], Nora Tramm[2,3†], Jarred Sanders[4†], Shachar Iwanir[2,3], Ian A Shirley[2,3], Erel Levine[5,6], David Biron[1,2,3]***

[1]Institute for Biophysical Dynamics, University of Chicago, Chicago, United States; [2]Department of Physics, University of Chicago, Chicago, United States; [3]James Franck Institute, University of Chicago, Chicago, United States; [4]Committee on Genetics, Genomics, and Systems Biology, University of Chicago, Chicago, United States; [5]Department of Physics, Harvard University, Cambridge, United States; [6]Center for Systems Biology, Harvard University, Cambridge, United States

**Abstract** Biological homeostasis invokes modulatory responses aimed at stabilizing internal conditions. Using tunable photo- and mechano-stimulation, we identified two distinct categories of homeostatic responses during the sleep-like state of *Caenorhabditis elegans* (lethargus). In the presence of weak or no stimuli, extended motion caused a subsequent extension of quiescence. The neuropeptide Y receptor homolog, NPR-1, and an inhibitory neuropeptide known to activate it, FLP-18, were required for this process. In the presence of strong stimuli, the correlations between motion and quiescence were disrupted for several minutes but homeostasis manifested as an overall elevation of the time spent in quiescence. This response to strong stimuli required the function of the DAF-16/FOXO transcription factor in neurons, but not that of NPR-1. Conversely, response to weak stimuli did not require the function of DAF-16/FOXO. These findings suggest that routine homeostatic stabilization of sleep may be distinct from homeostatic compensation following a strong disturbance.

**\*For correspondence:** david.
biron@gmail.com

†These authors contributed
equally to this work

**Competing interests:** The
authors declare that no
competing interests exist.

**Reviewing editor**: Ronald L
Calabrese, Emory University,
United States

## Introduction

Sleep architecture—the duration, timing, and order of individual stages of sleep—is derived from a combination of internal timekeeping pathways, a drive towards an appropriate baseline (sleep pressure), and external constraints. Collectively, the use of both mammalian and non-mammalian models has suggested that sleep is phylogenetically ancient and evolutionarily conserved (*Campbell and Tobler, 1984*; *Sehgal and Mignot, 2011*; *Nelson and Raizen, 2013*). The key behavioral hallmarks of sleep are episodic reduced motion, reversibility, typical postures, sensory gating, and homeostasis (*Campbell and Tobler, 1984*). Generally, the homeostatic drive underlies correlations between the strength and duration of a disruption and the subsequent duration and quality of sleep. Behavioral signatures of homeostasis include faster time-courses of wake-to-sleep transitions, prolonged periods of sleep, and increased arousal thresholds following a period of deprivation that increases sleep pressure (*Moses et al., 1975*; *Tobler, 1983*; *Hendricks et al., 2000*; *Allada and Siegel, 2008*; *Raizen et al., 2008*).

The nematode *Caenorhabditis elegans* is the simplest model organism that has been shown to exhibit a sleep-like state to date (*Raizen et al., 2008*; *Nelson and Raizen, 2013*; *Cho and Sternberg, 2014*). The 2–3 hr period of lethargus, a developmental stage that precedes the termination of each larval stage, is characterized by behavioral quiescence, a cessation of feeding, reduced or delayed

**eLife digest** The regenerative properties of sleep are required by all animals, with even the simplest animal, the nematode *Caenorhabditis elegans*, displaying a sleep-like state called lethargus. During development, nematodes must pass through four larval stages en route to adulthood, and the end of each stage is preceded by a period of lethargus lasting 2 to 3 hr.

Human sleep is divided into distinct stages that recur in a prescribed order throughout the night. Nematodes, on the other hand, simply experience alternating periods of activity and stillness as they sleep. Nevertheless, in both species, any disruptions to sleep automatically lead to adjustments of the rest of the sleep cycle to compensate for the disturbance and to ensure that the organism gets an adequate amount of sleep overall.

To date, it has been assumed that a single mechanism is responsible for adjusting the sleep cycle after any disturbance, regardless of its severity. However, Nagy, Tramm, Sanders et al. now show that this is not the case in *C. elegans*. Sleeping nematodes that were lightly disturbed by exposing them to light or to vibrations—causing them to briefly increase their activity levels—compensated for the disturbance by lengthening their next inactive period. By contrast, worms that were vigorously agitated by stronger vibrations showed a different response: the alternating pattern of stillness and activity was disrupted for several minutes, followed by an overall increase in the length of time spent in the stillness phase.

Experiments using genetically modified worms revealed that these two responses involve distinct molecular pathways. A signaling molecule called neuropeptide Y affects the response to minor sleep disruptions, whereas a transcription factor called DAF-16/FOXO is involved in the corresponding role after major disruptions. Given that neuropeptide Y has already been implicated in sleep regulation in humans and flies, it is not implausible that similar mechanisms may occur in response to disturbances of our own sleep.

responses to external stimuli, a distinct posture, and compensation following deprivation (*Van Buskirk and Sternberg, 2007*; *Raizen et al., 2008*; *Schwarz et al., 2012*; *Iwanir et al., 2013*; *Cho and Sternberg, 2014*). The *C. elegans* homolog of the circadian clock protein PERIOD is required for synchronization of lethargus, and its mRNA levels track the developmental/molting cycle (*Jeon et al., 1999*; *Allada et al., 2001*; *Tennessen et al., 2006*; *Monsalve et al., 2011*). Additional conserved signaling pathways that exhibit functional similarities in mammalian, insect, and nematode sleep include the epidermal growth factor (EGF) (*Kramer et al., 2001*; *Snodgrass-Belt et al., 2005*; *Foltenyi et al., 2007*; *Van Buskirk and Sternberg, 2007*; *Zimmerman et al., 2008*), the cyclic GMP-dependent protein kinase PKG (*Van Buskirk and Sternberg, 2007*; *Raizen et al., 2008*; *Langmesser et al., 2009*), cAMP-dependent signaling (*Hendricks et al., 2001*; *Graves et al., 2003*; *Raizen et al., 2008*), $G_s$ signaling, and genes acting downstream of dopamine signaling (*Singh et al., 2014*).

Homeostatic regulation within *C. elegans* lethargus was previously examined by manually depriving the animals of quiescence. After a deprivation period of 30 min during lethargus, the onset of long response latencies to chemical stimuli was accelerated. In addition, mechanical stimulation for 60 min at the time that the onset of lethargus was expected resulted in increased subsequent peak quiescence (*Raizen et al., 2008*). Recently, quiescent behavior and homeostatic rebound were also seen when a sleep-like state was induced anachronistically in adult animals, suggesting that developmental factors are not essential for neuromodulation during *C. elegans* sleep (*Cho and Sternberg, 2014*).

*C. elegans* can locomote forward or backward by propagating dorsoventral body bends from anterior to posterior or vice versa, respectively. Alternatively, they move in a variety of non-directional manners collectively referred to as dwelling (*Gray et al., 2005*; *von Stetina et al., 2006*; *Gallagher et al., 2013*; *Gjorgjieva et al., 2014*). During lethargus, *C. elegans* prominently exhibit quiescence—the complete absence of dynamic muscle contraction. Alternating bouts of locomotion and quiescence comprise the simple architecture of *C. elegans* sleep (*Raizen et al., 2008*; *Iwanir et al., 2013*). In a previous study, we have shown that the durations of these bouts are correlated (*Iwanir et al., 2013*), but the mechanisms underlying this process of routine stabilization were not examined. In this study, we analyze the behavioral responses of sleeping nematodes under undisturbed, weakly disturbed, and strongly disturbed conditions. To do so, we continuously assayed the locomotion of

*C. elegans* from the mid fourth intermolt stage (L4int), through the fourth lethargus stage (L4leth), and into the mid young adult stage (YA).

We found that weak photo- or mechano-stimulation transiently skewed the dynamics of bouts while preserving the characteristic pairwise correlations. Thus, under unperturbed or weakly perturbed conditions, homeostatic compensation manifested as a transient extension of quiescence bouts (and shortening of motion bouts under some conditions) in response to prolonged motion. This form of compensation under low noise conditions, termed micro-homeostasis (*Iwanir et al., 2013*; *Nelson and Raizen, 2013*), required the function of the neuropeptide Y (NPY) receptor homolog, NPR-1.

In contrast, strong stimuli induced a qualitatively different homeostatic response: the animals moved continuously for several minutes, after which quiescence monotonically returned to its baseline level. Compensation for the motion induced by a strong stimulus manifested as an upshift in the baseline fraction of time spent in quiescence, rather than a transient extension of quiescence bouts. The homeostatic responses to strong stimuli required the function of the DAF-16/FOXO in neurons (see also *Driver et al., 2013*) but not the function of NPR-1. Conversely, micro-homeostasis was not abolished in *daf-16* mutants.

In addition, we show that neuropeptidergic signaling is not strictly required for maintaining high levels of mean quiescence during lethargus. The loss of function of UNC-31/CAPS, a calcium-dependent activator protein required for dense core vesicle exocytosis (*Avery et al., 1993*; *Charlie et al., 2006*), resulted in a minor reduction of overall quiescence. In contrast, quiescence was strongly suppressed by the loss of the subsets of mature neuropeptides that were processed by the EGL-3 proprotein convertase or the EGL-21 carboxypeptidase E (CPE) (*Kass et al., 2001*; *Jacob and Kaplan, 2003*; *Husson et al., 2006, 2007*). As previously suggested (*Stawicki et al., 2013*), this apparent discrepancy can be resolved: collectively, our data indicate that a balance between inhibitory and excitatory contributions from different peptides modulates the duration of bouts of quiescence.

Our findings support a model in which locomotion during lethargus is coupled to a measure of increased sleep pressure. Quiescence serves to ameliorate this pressure and homeostatic regulation dynamically maintains an appropriate quiescence baseline. Interestingly, the homeostatic routine stabilization of motion and quiescence in low-noise environments is mechanistically distinct from homeostatic responses following strong, stressful, disruptions. To our knowledge, the analysis presented here is the first to identify this distinction.

## Results

### Motion plays a causal role in prolonging quiescence during lethargus

Homeostatic regulation of lethargus was previously examined using manually delivered strong mechanical stimuli, after which baseline levels of responsiveness were regained in 4 min (*Raizen et al., 2008*). However, even undisturbed animals compensate for spontaneous prolonged motion with prolonged quiescence during lethargus (*Iwanir et al., 2013*). Therefore, a mechanism that dynamically stabilizes lethargus behavior may be invoked by motion in quiet or weakly noisy environments. If so, weak stimuli should transiently skew the bout architecture by elongating motion bouts and causing a subsequent (compensatory) extension of quiescence bouts. To test this, we first exposed wild-type animals at the fourth intermolt larval stage, L4int, to pulses of blue light of intensities ranging from 0.3–100 mW/cm$^2$ and measured their responses using the frame subtraction method. In brief, this method consists of digitally recording the behavior of the animals and assessing the levels of motion and quiescence based on the number of pixels that change their brightness between consecutive frames (see *Nagy et al., 2014*). The observed responses depended on the light intensity and the duration of the stimulus, and we determined that a 15 s pulse of light at an intensity of 20–40 mW/cm$^2$ evoked weak, reproducible responses (*Figure 1—figure supplements 1,2*). Interestingly, we noted that 5 s pulses failed to produce a sharp response specifically during lethargus. This suggested that the animals were less responsive during lethargus and that reduced responsiveness could be assayed separately from delayed responsiveness (*Raizen et al., 2008*).

The response of L4int larvae and post lethargus young adult (YA) animals to weak blue light stimuli consisted of elevated levels of locomotion, which persisted for 15–25 s after the end of the pulse, followed by a 2 min decline back to baseline locomotion levels. In contrast, during L4leth the average level of locomotion crossed its baseline 1 min after it peaked, proceeded to fall below it for 2–3 additional min (p < 0.01), and only then stabilized at baseline levels (*Figure 1A*). The transient trough in

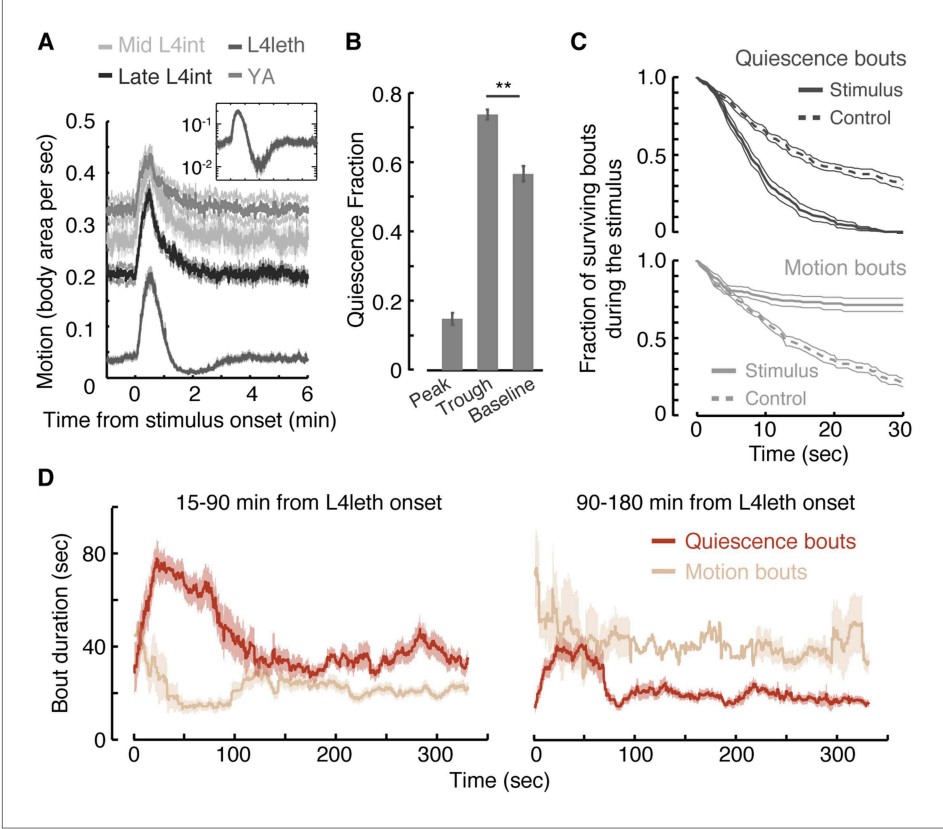

Figure 1. Motion plays a causal role in determining the duration of subsequent quiescence during lethargus. (A) Wild-type animals at the mid L4int, late L4int, L4leth, and YA stages were exposed to 30 s light stimuli at an intensity of 20 mW/cm². All stimuli were initiated at t = 0. Outside lethargus, locomotion monotonically decayed to baseline levels in 2 min. During lethargus, the peak in locomotion was followed by a trough prior to returning to baseline. Insets: the responses during lethargus shown on a semi-log scale. (B) The fractions of quiescence were calculated for 1 min intervals centered at the times of the peak and trough of the L4leth responses, as well as for their respective pre-stimulus baselines. Plots and bars depict mean ± s.e.m obtained from datasets of N = 40–50 animals per condition. Asterisks indicate p < 0.001. (C) Survival curves of quiescence and motion bouts of wild-type animals exposed to a 30 s, 20 mW/cm², blue light stimulus during the first hour of L4leth. Bouts were identified using the frame subtraction method and control data were obtained from the same animals, but 8 min after the stimulus (non-stimulated control animals were also assayed, analyzed the same way, and found to be indistinguishable from this control group). Mean ± s.e.m, N > 200 bouts for each condition. (D) The dynamics of bouts obtained from a posture-based analysis following a 15 s, 20 mW/cm², blue light stimulus. Left and right panels correspond to the first and second halves of L4leth, respectively. See also Figure 1—figure supplements 1–3. Plots depict mean ± s.e.m, smoothed using a 30 s running window average. N = 40 animals.

The following figure supplements are available for figure 1:

Figure supplement 1. Calibration of weak blue light stimuli.

Figure supplement 2. Responses to weak light stimuli.

Figure supplement 3. Responses during quiescence and motion.

---

locomotion resulted from an increase in the fraction of time the animals were quiescent rather than from slower motion (Figure 1B).

The presence of a weak light stimulus terminated bouts of quiescence prematurely and extended bouts of motion (Figure 1C, p < 0.01 in both cases). Identical responses were observed whether the onset of the stimulus interrupted a bout of quiescence or motion (Figure 1—figure supplement 3). The increase in the fraction of time spent in quiescence after the stimulus was removed could have

been caused by an extension of quiescence bouts, shortening of motion bouts, or both. To distinguish between these possibilities, we turned to an accurate and computationally intensive behavioral analysis. This previously described approach was based on continuous measurements of the dynamics of body posture at high temporal and spatial resolutions (*Iwanir et al., 2013*; *Nagy et al., 2014*). Using this analysis and a 15 s weak light stimulus, we measured the durations of bouts of motion and quiescence after the stimulus was turned off. We found that, during the first half of lethargus, the compensatory response was comprised of an increase and a decrease in the durations of quiescence and motion bouts, respectively (*Figure 1D*, p < 0.01). During the second half of lethargus, a compensatory increase in the durations of quiescence bouts was still observed. Taken together, these findings revealed that motion during non- or weakly-interrupted lethargus, but not during the L4int or YA stages, caused a compensatory transient increase in quiescence.

## The character of behavior during a motion bout affects subsequent quiescence

Posture-based analysis allowed for improved measurements of pairwise correlations between durations of bouts of motion and subsequent bouts of quiescence in undisturbed animals (*Figure 2A*, R = 0.47 ± 0.03, p < 0.001) (*Iwanir et al., 2013*). This approach revealed that these correlations gradually decayed as lethargus progressed (*Figure 2A*). Moreover, it enabled us to compare groups of motion bouts that contained different qualities of motion despite having similar overall durations. We could thus address the question of whether vigorous or directed motion in and of itself might affect the subsequent bout of quiescence.

To compare between groups of motion bouts of equal durations, we binned the bouts recorded during the first 90 min of L4leth of non-stimulated wild-type animals in 2 s wide bins. For each bin, we calculated the median vigor of locomotion as measured by the rate of change of body-curvature (*Figure 2B*, left panel). The motion bouts (from each bin) were then separated into two groups: those exhibiting higher-than-median or lower-than-median vigor with respect to their bin of origin. Each of the two groups therefore contained bouts of all durations and, importantly, the average duration of a motion bout was the same in both groups (*Figure 2B*, middle panel). Having controlled for the mere durations, we found a significant effect of the level of locomotion on the duration of the subsequent quiescence bout (*Figure 2B*, right panel). A similar analysis, performed exclusively on bouts that contained directed motion, considered the separation between the two groups based on the fraction of the bout spent in directed motion and produced similar results (*Figure 2C*). We thus conclude that enhanced or directed locomotion during a bout of a given duration positively affects subsequent quiescence.

## Behavioral responses to external stimuli are distinct during lethargus

Dwelling behavior during a motion bout appears similar to dwelling behavior outside of lethargus. However, although locomotory responses were shown to be delayed during lethargus (*Raizen et al., 2008*; *Cho and Sternberg, 2014*), they were not previously examined in detail. We asked whether responses to a weak stimulus during a bout of motion were distinct from responses during a bout of quiescence, from responses outside of lethargus, or from both.

To examine behavioral responses to external perturbations throughout this study, we used a recurrent stimulus assay: animals were repeatedly exposed to a stimulation regime of brief, widely spaced, photo- or mechano-stimuli. The duration of each individual stimulus was 0.4 or 15 s, depending on the type of assay, and the spacing between consecutive stimuli was 15 min. Animals were continuously assayed for 10 hr from the L4int stage to the mid YA stage. A diagram outlining the design of these assays is depicted in *Figure 3A*. Since multiple 15-min cycles were aligned and averaged, the resulting data had periodic boundaries. For instance, the same 1-min period could be referred to as the 15th minute after the stimulus or the 1 min just prior to the stimulus (see *Figure 3—figure supplement 1*).

In our hands, during lethargus, the transient changes in behavioral dynamics that constituted the short-term response to a stimulus were limited to a period of 3 min immediately following stimulation. After this short-term response was complete, behavioral dynamics returned to a steady state characteristic of the conditions of the experiment. Consequently, baseline behavior for each set of experimental conditions was defined as the steady state measured during a 5 min period starting 10 min after a stimulus and 5 min prior to the subsequent stimulus (labeled explicitly in *Figure 3A* and *Figure 3—figure supplement 1*, and depicted as t = −5…0 min in subsequent panels).

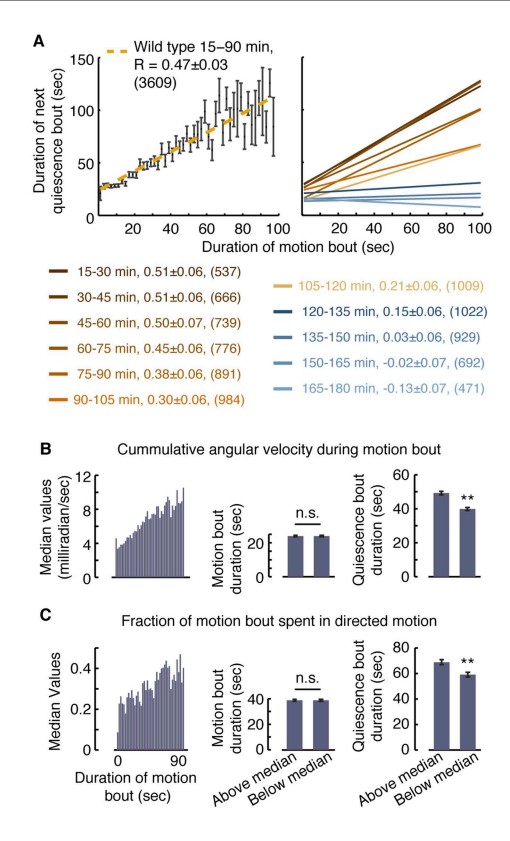

**Figure 2**. Vigorous or directed motion extends the duration of subsequent quiescence during lethargus. (**A**) Posture-based analysis improved the measurement of pairwise correlations between the durations of motion bouts and those of subsequent quiescence bouts in undisturbed wild-type animals (R = 0.47 ± 0.03, N = 3609 bouts from 40 animals, p < 0.05). As a guide to the eye, motion bouts were grouped according to their durations in 2 s wide bins. The mean ± s.e.m duration of the subsequent quiescence bouts for each bin was plotted and these mean values were fitted to a line. In addition, pairwise correlation coefficients were calculated for each 15 min interval of L4leth separately. As a guide to the eye, linear fits to the binned data are depicted. In all cases, the errors were defined as the 95% confidence intervals and the number of bouts is given in parentheses. (**B–C**) The overall levels of motion (**B**) and the fraction of directed motion (**C**) during a motion bout have a significant effect on subsequent quiescence. Overall motion was defined as the mean time derivative of the absolute values of 18 angles along the body and directed motion was defined as either forward or backward locomotion, as opposed to dwelling (**Nagy et al., 2014**). Left: the median values of the overall vigor of motion (**B**) and the fraction of directed motion (**C**) as a function of the duration of the motion bouts (binned in 2 s bins). Middle (right): the durations of motion (quiescence) bouts calculated separately for the group of bouts that was above

*Figure 2. Continued on next page*

Outside of lethargus, the onset of a weak light stimulus evoked a sharp rise in the propensity for forward locomotion, while backward locomotion was suppressed (L4int) or unchanged (YA). From the time of the offset of the stimulus, forward locomotion monotonically declined and reversals returned to baseline levels (L4int) or were briefly elevated (YA). The probability of forward locomotion decayed to its baseline value in 3 min as the baseline balance between directed motion and dwelling was re-established (**Figure 3B** left and right panels).

During L4leth, the onset of the stimulus evoked a sharp rise in the propensities for both forward and backward locomotion. The offset of the stimulus did not reverse the increasing propensity for moving forward. Rather, forward locomotion persisted for 20 s after the light was turned off and subsequently fell below its steady state value while quiescence levels exceeded their baseline (**Figure 3B** middle panel). Similar features were observed for responses throughout L4leth (**Figure 3C**), and regardless of whether the onset of the stimulus occurred during a motion or a quiescence bout (data not shown). Thus, responses to weak stimuli revealed similar locomotory responses during bouts of motion and quiescence and differentiated both types of bouts from the L4int and YA stages.

## Homeostatic responses to weak and strong stimuli are distinct

The compensatory extension of quiescence bouts after a weak light stimulus was distinct from previously reported responses to manually delivered strong mechanical stimuli (**Raizen et al., 2008**; **Driver et al., 2013**). To test whether the modulation of bout duration was specifically evoked by light, we assayed animals that were exposed to a mechanical stimulus: vibrations at a frequency of 1 kHz (**Nagy et al., 2014**). The strength of the stimulus was tuned by varying its duration. Outside lethargus, a 0.4 s stimulus elicited a transient increase in reversals followed by a brief enhancement of the propensity for forward locomotion, while a 15 s stimulus elicited a similar initial recoil followed by an enhancement of forward locomotion that lasted for 10 min (**Figure 4A,B**, left). We thus refer to the short stimulus as weak and the longer stimulus as strong.

During L4leth, weak mechanical stimuli induced transient backward locomotion, followed by enhanced quiescence. Specifically, the first bout of quiescence after the recoil was elongated, and the architecture of locomotion and quiescence returned to baseline 1 min after the stimulus was

*Figure 2. Continued*

or below the median of its respective bin. The durations of quiescence bouts differed significantly between the two groups. N = 40 animals, error bars depict s.e.m, p < 0.01.

delivered (***Figure 4A***, right). In contrast, the strong stimulus disrupted the architecture of behavior during lethargus: it was followed by several minutes of enhanced motion and a monotonous relaxation to baseline quiescence levels. Upon return to baseline, quiescence bouts were not transiently extended such that a peak in quiescence was not observed. However, compensation took on a different form: the overall level of baseline quiescence was elevated (***Figure 4B,C***). This overall elevation of quiescence was consistent with previously reported compensation after strong stimulation (***Raizen et al., 2008***; ***Driver et al., 2013***). These results suggested that there were two regimes of disruption and compensation. Weak perturbations resulted in a transient modulation of bout durations that did not disrupt (and could even enhance) the characteristic correlations of the bouts architecture. In contrast, a strong perturbation abrogated the routine dynamics of bouts for several minutes and increased the baseline fraction of quiescence thereafter.

## The neuropeptide Y receptor homolog, NPR-1, plays a role in modulating quiescence in both unperturbed and weakly stimulated animals

Neuropeptide Y (NPY) and its receptors have been implicated in the regulation of sleep (albeit in different manners) in humans, rats, fruit flies, and nematodes (***Antonijevic et al., 2000***; ***Tóth et al., 2007***; ***Dyzma et al., 2010***; ***Van den Pol, 2012***; ***Choi et al., 2013***; ***Nagy et al., 2014***). To test their role in mediating micro-homeostasis, we assayed animals carrying two mutant alleles of the *C. elegans* NPY receptor homolog gene, *npr-1*. In our hands, overall quiescence in animals carrying the *npr-1(ky13)* allele, a glutamine to ochre nonsense mutation at codon 61, was only mildly different from wild-type (***Nagy et al., 2014***). However, bout correlations in these mutants were significantly reduced. The *npr-1(ad609)* allele induced similarly reduced bout correlations and a more pronounced defect in the durations of quiescence bouts throughout lethargus (***Figure 5A***).

We next assayed the responses of *npr-1* mutants to weak blue light stimuli. Using the frame subtraction method, we could not detect significant compensation following the excess motion induced by the stimulus in either of the two mutant strains (***Figure 5B***). The posture-based analysis confirmed their severe defect in modulation of bout durations (***Figure 5C,D***). The overall activity and, in particular, the initial response of *npr-1* mutants to the weak stimulus were similar to wild-type. This indicated that the mutants were not defective in sensing the stimulus or in their locomotory capabilities but specifically in their ability to compensate for a weak disturbance. In contrast, *npr-1* mutants exhibited wild-type-like compensation following strong mechanical stimuli: when animals carrying either of the two mutant alleles were exposed to a strong mechanical stimulus, their baseline fraction of quiescence was elevated as compared to non stimulated or weakly stimulated animals (***Figure 5E***). Thus, in addition to the phenotypic differences described above, homeostatic compensation during undisturbed or weakly disturbed lethargus was affected by NPR-1, while homeostatic compensation for strong stimuli was not. We concluded that the routine stabilization of lethargus behavior in low-noise environments and the homeostatic compensation for stressful disturbances were mechanistically separable. These findings are consistent with a model in which NPR-1 modulates quiescence during lethargus in response to spontaneous or induced mild variations in locomotion.

## Peptidergic signaling is required for micro-homeostasis

NPR-1 is a predicted neuropeptide receptor and the FMRFamide-like neuropeptides encoded by *flp-18* and *flp-21* were shown to be two of its ligands. (***De Bono and Bargmann, 1998***; ***Kubiak et al., 2003***; ***Rogers et al., 2003***; ***Kim and Li, 2004***). We therefore asked whether peptidergic release from dense core vesicles (DCVs) was required for micro-homeostasis. To answer this question, we assayed the loss of function of UNC-31, the sole *C. elegans* ortholog of mammalian calcium-dependent activator protein for secretion (CAPS) required for DCV exocytosis (***Avery et al., 1993***; ***Charlie et al., 2006***). To confirm that the observed phenotype was explained by the mutation of interest, we tested a strong loss of function allele, *unc-31(e169)*, and a putative null allele, *unc-31(e928)*, (***Charlie et al., 2006***; ***Speese et al., 2007***).

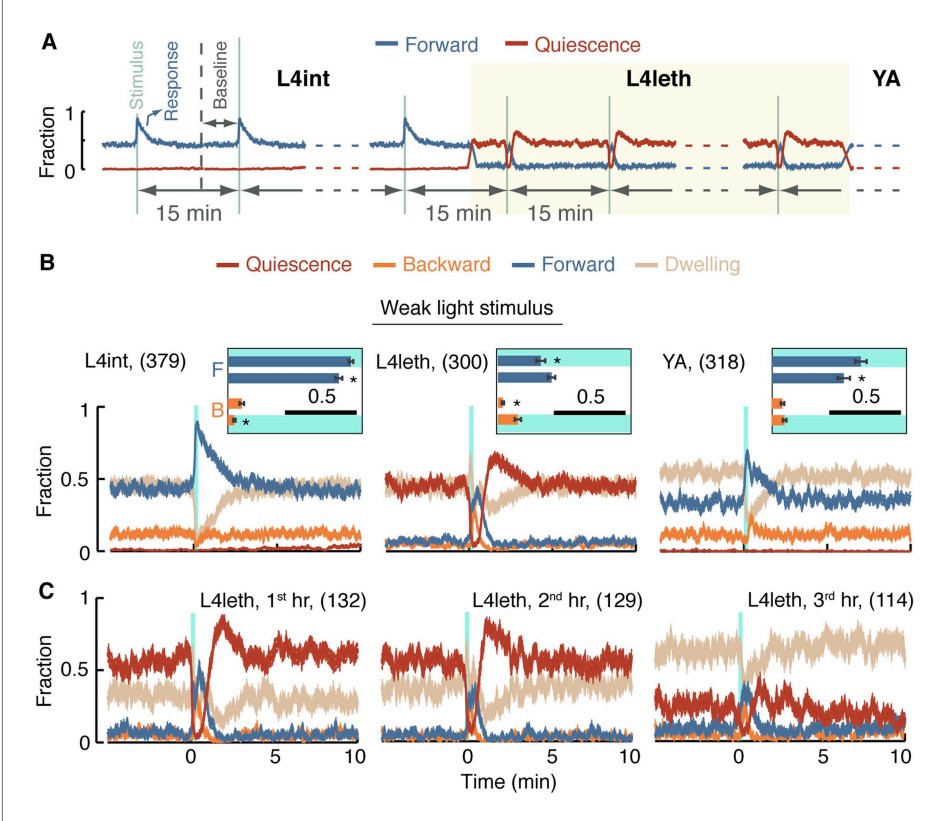

**Figure 3**. A posture-based analysis of locomotion responses to weak light stimuli. (**A**) A diagram describing the repeated stimulus assay, in which a generic brief stimulus (vertical lines) was repeatedly delivered at 15 min intervals (long horizontal arrows). Each assay started at the mid L4int stage, continuously progressed through L4leth (shaded area), and ended at the mid YA stage. For the purpose of illustration, the blue and red lines symbolize tentative probabilities of forward locomotion and quiescence, respectively. Baseline behavior was measured during the 5-min period starting 10 min after a stimulus, or equivalently, 5 min prior to the subsequent stimulus. The beginning of the first baseline period is depicted by a dashed vertical line. (**B**) The fraction of forward locomotion, backward locomotion, dwelling, and quiescence before, during, and after a weak (15 s, 20 mW/cm² blue light) stimulus provided at the L4int (left), L4leth (middle), and YA (right) stages. A compensatory post-stimulus enhancement of quiescence, as well as enhanced reversals during the stimulus, and a rising propensity for forward locomotion after the stimulus was turned off were uniquely observed during lethargus. Insets: the fraction of forward locomotion before and after the offset of the stimulus (top) and the fraction of backward locomotion before and after the onset of the stimulus. Shading denotes the presence of the light stimulus. All fractions were calculated from the 7.5-s period (half of the duration of the stimulus), the scale bars represent a fraction of 0.5, and asterisks denote p < 0.05. (**C**) The data from the middle panel of (**B**) plotted separately for the first, second, and third hours of L4leth. Enhanced quiescence was observed in all three cases, although it was less prominent during the third hour. Plots in panels (**B**, **C**) depict mean ± s.e.m and the number of stimuli assayed is noted in parentheses for each condition.

The following figure supplement is available for figure 3:

**Figure supplement 1**. The averaged behavior data have periodic boundaries.

---

Under undisturbed conditions, the quiescence bouts of *unc-31* mutants were shorter than wild-type, but the overall amount of quiescence was only weakly reduced in these mutants (***Figure 6A,B***). Moreover, *unc-31* mutants did not exhibit paralysis or anachronistic quiescence outside of lethargus and their overall locomotory behavior during lethargus was similar to wild-type (***Nagy et al., 2013*** and data not shown). Nevertheless, pairwise correlations between subsequent bouts in these mutants were abolished (***Figure 6B***). This could indicate that the absence of a group of functional neuropeptides impaired the dynamic extension of quiescence bouts in response to variations in durations and compositions of motion bouts. Alternatively, the quiescence bouts of *unc-31* mutants may be too short

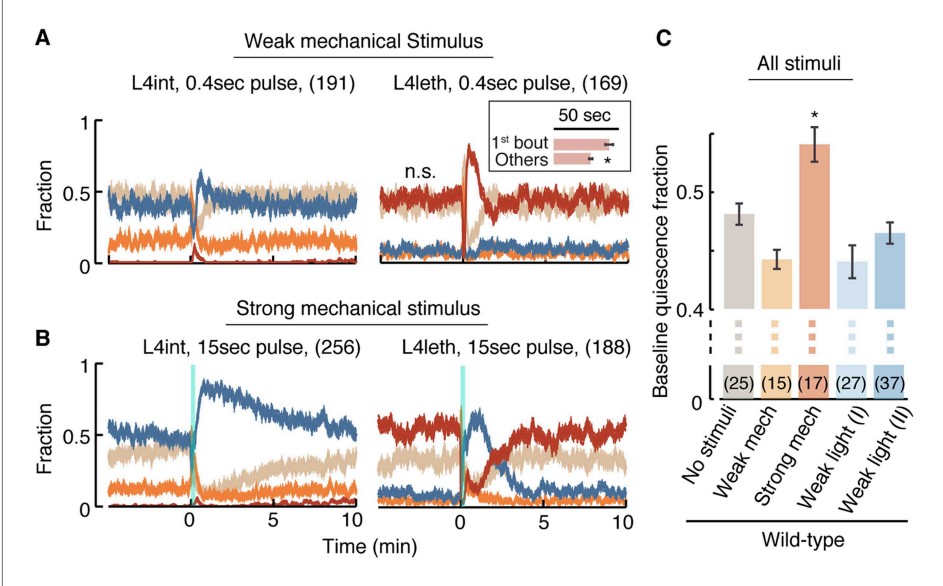

**Figure 4**. A posture-based analysis of locomotion responses to weak and strong mechanical stimuli. (**A**) A weak mechanical stimulus (0.4 s of 1 kHz vibrations) produced a reversal followed by a small elevation of forward locomotion in L4int larvae (left) and a brief reversal followed by enhanced quiescence during L4leth (right). Inset: the first quiescence bout after the stimulus was longer than subsequent bouts (p < 0.05). (**B**) A strong mechanical stimulus (15 s of 1 kHz vibrations) produced reversals followed by a prolonged (10 min) elevation of forward locomotion in L4int larvae (left) and a brief reversal followed by elevated levels of directed motion for 4–5 min during L4leth (right). Notably, quiescence returned to its baseline value without transiently exceeding it. (**C**) Mean baseline fraction of quiescence was measured during the baseline period (see *Figure 3A*). The baseline fraction of quiescence was significantly higher in strongly stimulated animals as compared to unstimulated and weakly stimulated animals. Weak light I and II labels refer to stimulus strengths of 20 and 40 mW/cm² blue light, respectively. Plots in panels (**A**, **B**) depict mean ± s.e.m and the error bars in panel (**C**) depict ±s.e.m and asterisks denote p < 0.05. The number of stimuli assayed is noted in parentheses for each condition.

to sustain detectable correlations. We favor the first explanation for two reasons. First, the Hawaiian strain (a wild isolate of *C. elegans*) exhibited quiescence bouts that were comparable in duration to those of *unc-31* mutants but nevertheless maintained wild-type correlations during minutes 45–120 from the onset of L4leth (*Figure 6—figure supplements 1,2*). Second, when bout pairs containing longer quiescence bouts were excluded from the wild-type dataset, such that the mean duration of the remaining quiescence bouts equaled that of *unc-31* mutants, the pairwise correlation between the remaining bouts was reduced to R = 0.2 ± 0.03, but not abolished.

We next assayed the responses of *unc-31* mutants to weak (light) stimuli. Animals carrying the *unc-31(e928)* null mutation, as well as animals carrying the *unc-31(e169)* loss of function mutation, exhibited a diminished ability to prolong quiescence bouts in response to prolonged motion. The stronger defect was observed in *unc-31(e928)* mutants (*Figure 6C,D*). Collectively, these findings suggest that peptidergic signaling plays a key role in regulating micro-homeostasis.

In addition to *unc-31*, we assayed mutants in which neuropeptide processing was disrupted due to the loss of function of: (i) the proprotein convertase required for preprocessing of many, but not all, neuropeptides, EGL-3 (*Kass et al., 2001*; *Husson et al., 2006*), or (ii) the carboxypeptidase E (CPE) required to complete the processing of the majority of non insulin-like neuropeptides, EGL-21 (*Jacob and Kaplan, 2003*; *Husson et al., 2007*). Consistent with previous reports (*Turek et al., 2013*), overall quiescence during lethargus was significantly reduced in *egl-3* mutants, individual quiescence bouts were very short, and (as expected) correlations between bout durations were abolished. The loss of function of EGL-21 resulted in an identical phenotype, demonstrating that the phenotype was caused by the mutations of interest (*Figure 6A,B*). These results stood in contrast to the mild change in overall quiescence observed in *unc-31* mutants, and this apparent discrepancy is discussed below.

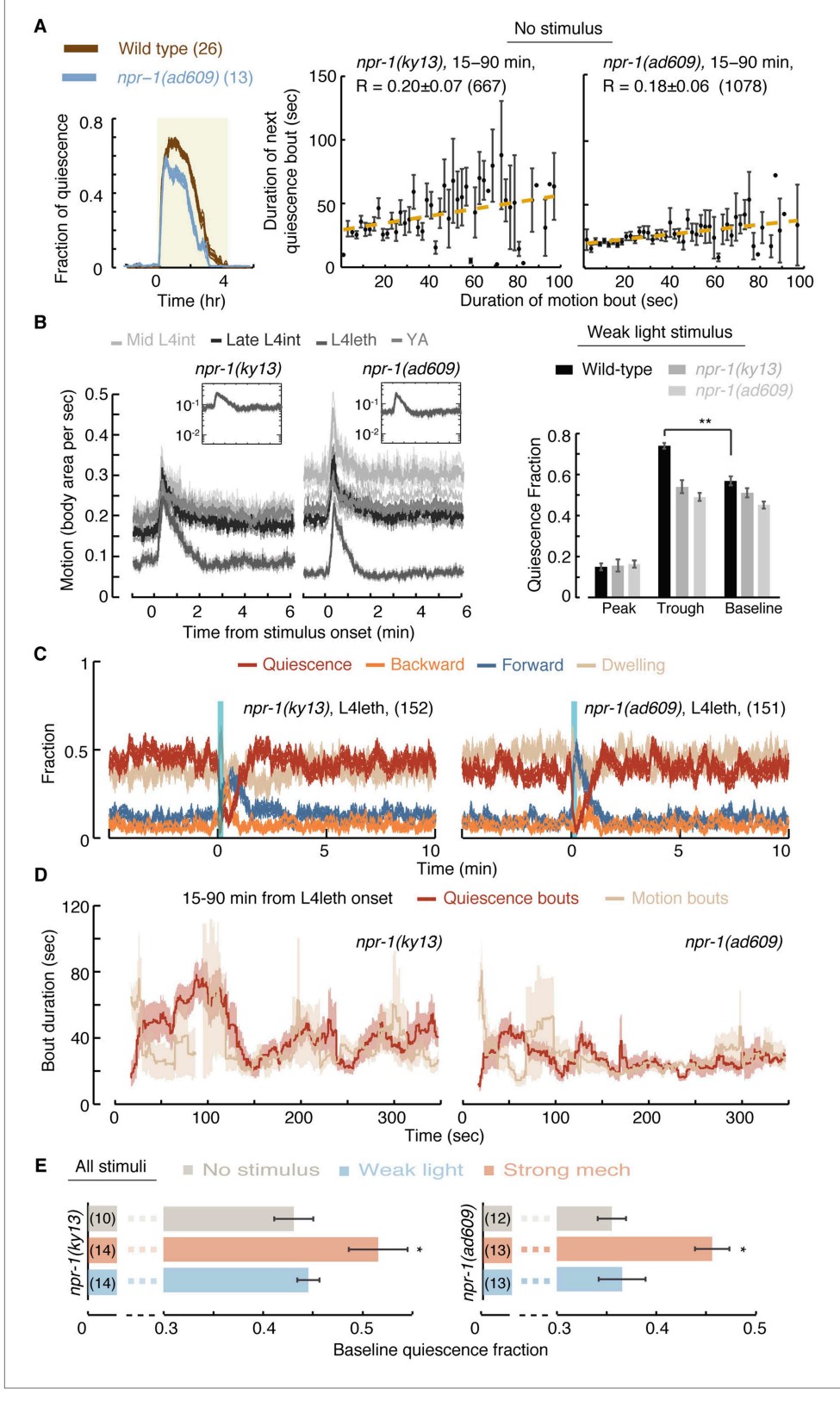

**Figure 5**. NPR-1 is required for micro-homeostasis but not for homeostatic responses to strong stimuli. (**A**) Undisturbed behavior of *npr-1* mutants. Left: the fraction of quiescence of wild-type animals and *npr-1(ad609)* mutants during L4leth (shaded area). The fraction of quiescence of *npr-1* mutants was recently

*Figure 5. Continued on next page*

*Figure 5. Continued*

published (*Nagy et al., 2014*) and plotted here for comparison. Plots depict mean ± s.e.m, the numbers of animals assayed are denoted in parentheses. Middle and right: pairwise bout correlations and plots of binned bouts (see *Figure 2A* for details). Pairwise correlations were significantly reduced in *npr-1* mutants (p < 0.05). All correlations are given with 95% confidence intervals and error bars depict ±s.e.m. The number of bouts in each case is denoted in parentheses. (**B**) L4int, late L4int, L4leth, and YA *npr-1* mutants were exposed to weak (15 s, 20 mW/cm² light) stimuli. All stimuli were initiated at t = 0. In *npr-1* mutants assayed using frame subtraction, a trough did not follow the transient increase in locomotion before returning to baseline. Insets: the responses during lethargus shown on a semi-log scale. For each strain, the quiescence fraction was calculated during 1 min intervals centered at the times of the peak and trough of the L4leth responses, as well as for their respective pre-stimulus baselines. Quiescence was not enhanced following the peak in locomotion in *npr-1* mutants. Plots and bars depict mean ± s.e.m obtained from datasets of N = 50–60 animals per condition. Asterisks and double asterisks denote p < 0.05 and p < 0.01, respectively. (**C**) A posture-based analysis of behavior of L4leth *npr-1* mutants: the fraction of forward locomotion, backward locomotion, dwelling, and quiescence before, during, and after a weak (15 s, 20 mW/cm², blue light) stimulus. The data were aligned by the time of the onset of the stimulus and then averaged. Plots depict mean ± s.e.m. In agreement with the frame subtraction measurements, the compensatory enhancement of quiescence fraction shortly after the stimulus was nearly abolished in *npr-1* mutants. N = 14 and 13 animals (*ky13* and *ad609*). (**D**) A posture-based analysis of bout dynamics of *npr-1* mutants following a weak stimulus. N = 14 and 13 animals, plots depict mean ± s.e.m, smoothed using a 30 s running window. (**E**) The mean baseline fractions of quiescence during the 5 min intervals prior to each stimulus tested. Similar to wild-type, baseline quiescence fraction was significantly higher in strongly stimulated animals as compared to non-stimulated and weakly stimulated *npr-1* mutants. See also *Figure 1—figure supplements 1–3*. Error bars depict ±s.e.m and asterisks denote p < 0.05. The number of stimuli assayed is noted in parentheses for each condition.

## The FMRFamide-like peptide FLP-18 plays a role in micro-homeostasis

The FMRFamide-related neuropeptides FLP-18 and FLP-21 were shown to be ligands of NPR-1, as well as two additional receptors (*Rogers et al., 2003*; *Cohen et al., 2009*). In addition, FLP-18 (but not FLP-21) was shown to act synergistically, in an inhibitory fashion, in the homeostatic response to motoneuron imbalance (see discussion and *Stawicki et al., 2013*). In our hands, *flp-21* mutants did not exhibit defective micro-homeostasis. We used posture analysis to assay unperturbed *flp-18(gk3036)* mutants and the frame subtraction method to assay *flp-18(gk3036)* and *flp-18(db99)* mutants in the presence of weak perturbations (*Cohen et al., 2009*). The overall quiescence fraction and the durations of quiescence bouts of *flp-18* mutants were comparable to those of *npr-1* mutants (*Figure 7A* and data not shown). However, the correlations between subsequent bouts in undisturbed *flp-18(gk3036)* mutants were intermediate between the wild-type and *npr-1* values: 0.33 ± 0.06, 0.47 ± 0.03, and 0.20 ± 0.07, respectively (p < 0.05, *Figure 7B*). When stimulated with blue light, both *flp-18* alleles were associated with defective compensatory responses, and the defect was more pronounced in *flp-18(db99)* mutants (*Figure 7C,D*).

If FLP-18 plays a role in micro-homeostasis then its production, secretion, or both may be temporally correlated with lethargus. To test this, we examined the temporal dynamics of expression during the L4int and L4leth stages of the *Pflp-18::flp-18::SL2::gfp* reporter, which contains the upstream promoter region and the entire genomic locus of *flp-18* (*Cohen et al., 2009*; *Stawicki et al., 2013*). As previously reported, expression was observed in several head and ventral cord (VC) neurons. We measured the total GFP fluorescence in head or VC neurons separately. Expression in head neurons was constant prior to the onset of and during L4leth (*Figure 7—figure supplement 1*). Surprisingly, the reporter expression in VC neurons differed between two sub-populations of animals. When low levels of fluorescence were initially detected during L4int, reporter fluorescence was enhanced more than twofold during the first half of L4leth. In contrast, initially high fluorescence levels were not further enhanced. Expression levels of the reporter in the VC neurons of the two sub-populations were similar during the second half of L4leth (*Figure 7—figure supplement 1*). The absence of a peak in fluorescence during lethargus in the initially strongly fluorescent sub-population may have resulted from non-physiological effects of overexpression. Alternatively, it may be the case that the shift in *flp-18* expression or secretion can precede the onset of lethargus or be conditioned on ambient levels during late L4int.

Since *Pflp-18::flp-18::SL2::gfp* expression peaked during the first half of lethargus, we examined the timing of the defect in bout correlations in *flp-18* mutants with respect to the onset of lethargus.

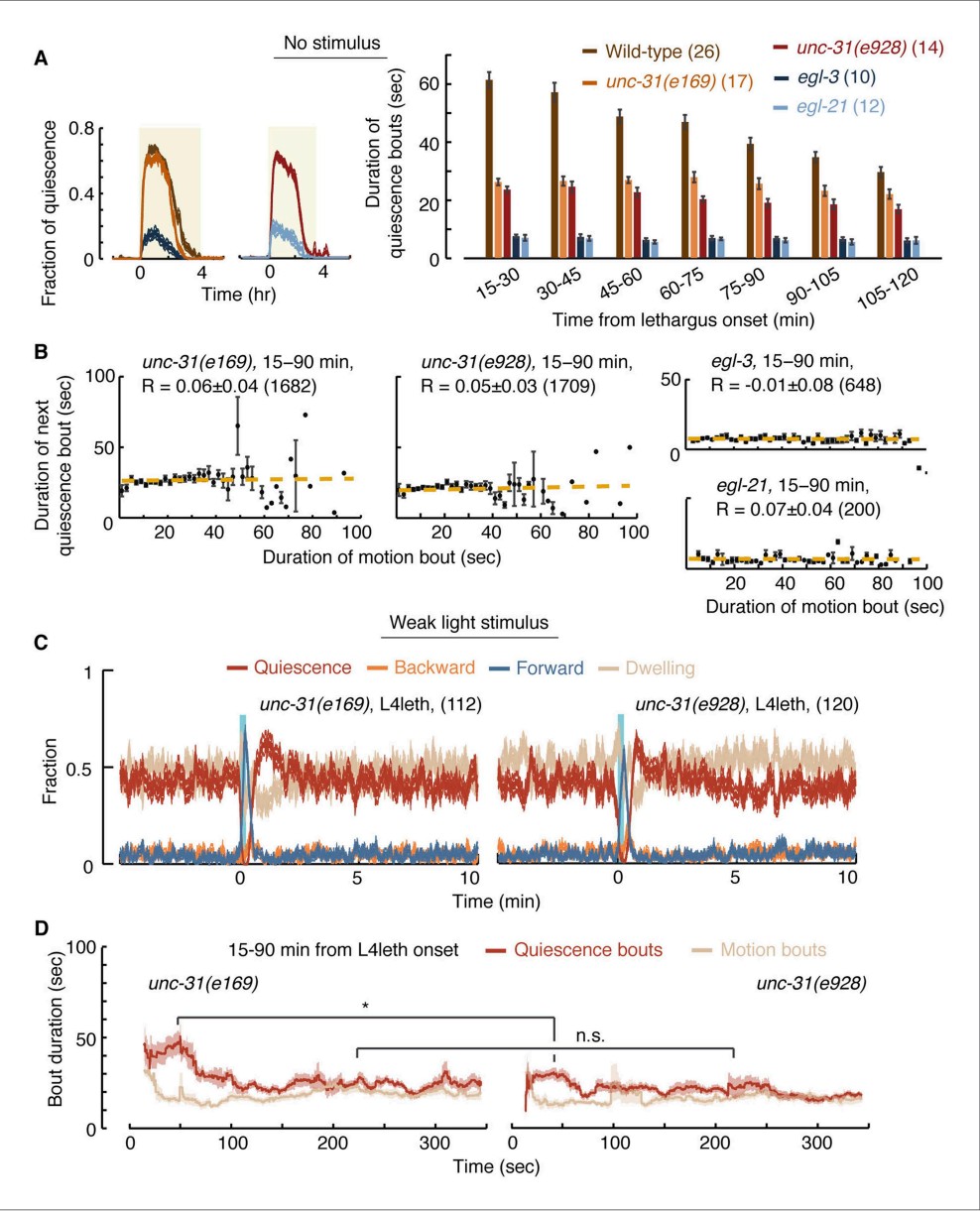

**Figure 6**. UNC-31/CAPS is not required for establishing a high fraction of quiescence during lethargus but is required for micro-homeostasis. (**A**) Left: the fraction of quiescence of wild-type animals and *unc-31*, *egl-3*, and *egl-21* mutants during L4leth (shaded area). Quiescence was strongly reduced by the loss of function of EGL-3 or EGL-21, but not UNC-31. Right: the mean durations of bouts of quiescence of the same wild-type and mutant animals during the 15-min period of L4leth. Plots and bars depict mean ± s.e.m, the numbers of animals assayed are denoted in parentheses. (**B**) Pairwise bout correlations and plots of binned bouts in undisturbed animals (see *Figure 2A* for details). Pairwise correlations were abolished in *unc-31*, *egl-3*, and *egl-21* mutants. All correlations are given with 95% confidence intervals (p < 0.05) and error bars depict ±s.e.m. The number of bouts in each case is denoted in parentheses. (**C**) A posture-based analysis of behavior of L4leth *unc-31* mutants: the fraction of forward locomotion, backward locomotion, dwelling, and quiescence before, during, and after a weak (15 s, 20 mW/cm², blue light) stimulus. See also *Figure 6—figure supplements 1,2*. (**D**) A posture-based analysis of bout dynamics of *unc-31* mutants following a weak stimulus. The duration of the motion induced by the weak stimulus was shorter than that of wild-type animals, and the compensatory enhancement of quiescence was weaker. N = 11 and 12 animals (*e169* and *e928*), plots depict mean ± s.e.m, smoothed using a 30 s running window.

*Figure 6. Continued on next page*

*Figure 6. Continued*
The following figure supplements are available for figure 6:
**Figure supplement 1**. Micro-homeostasis in undisturbed Hawaiian wild-isolates.
**Figure supplement 2**. Micro-homeostasis in undisturbed Hawaiian wild-isolates.

The positive pairwise bout correlations in *flp-18* mutants were found to be smaller than wild-type during the first hour of lethargus, but not during the second hour, corresponding to the observed period of upregulation in expression of the reporter (*Figure 7—figure supplement 2*). Collectively, these findings suggest that both FLP-18 and its known receptor, NPR-1, regulate micro-homeostasis during lethargus.

## Homeostatic responses to strong stimuli and micro-homeostasis are differentially regulated

Prolonged and stressful deprivation of quiescence during lethargus causes the translocation of DAF-16, a FOXO transcription factor that activates stress responses, into the nucleus. Moreover, *daf-16* mutants were shown to be defective in their behavioral response to prolonged deprivation (*Lin et al., 1997*; *Henderson and Johnson, 2001*; *Driver et al., 2013*). Although micro-homeostasis responses occur on a timescale that is too short to be regulated by changes in transcription, repeated weak stimuli may still be stressful. To test the roles of DAF-16 in regulating homeostasis during lethargus, we assayed *daf-16(mu86)* (*Libina et al., 2003*) mutants under no-, weak-, and strong-stimulus conditions. These mutants were similar to wild-type in their total fraction of quiescence, their initial responses to weak stimuli and subsequent compensation, their responses outside of lethargus to weak and to strong stimuli, and their initial responses during lethargus to strong stimuli. When not disturbed, the quiescence bouts of *daf-16* mutants were shorter than wild-type (data not shown) and their pairwise correlations between subsequent bouts were smaller, but not abolished (*Figure 8A–C*). A second mutant allele, *daf-16(mgDf50)* (*Ogg et al., 1997*), exhibited similar behavior under unstimulated conditions (data not shown). Thus, micro-homeostasis during *C. elegans* lethargus was mostly independent of DAF-16/FOXO signaling.

In contrast, the homeostatic compensation in our strong stimulus assay was completely abolished in both *daf-16* mutants (*Figure 8D*). To test where the function of *daf-16* was required, the function of *daf-16* was rescued under the control of the daf-16 native promoter (P*daf-16*), a pan-neuronal promoter (P*unc-119*), and a body-wall muscle promoter (P*myo-3*) (*Driver et al., 2013*). Homeostatic compensation for strong disturbances was restored when *daf-16* was expressed under its native promoter or in neurons, but not in muscles (*Figure 7D*). These findings differ from the reported role of DAF-16 in sleep homeostasis, assayed using response latencies to a noxious chemical, where rescue in muscles but not in neurons restored wild-type-like latencies (*Driver et al., 2013*). However, adult locomotion quiescence in *daf-2* mutants (an insulin/IGF-1 receptor homolog) was dependent on the function of DAF-16 in neurons (*Gaglia and Kenyon, 2009*). Thus, DAF-16 may act in multiple tissues to regulate different aspects of the homeostatic response in *C. elegans* sleep.

We noted that the baseline level of quiescence in undisturbed animals varied between the different transgenic strains (*Figure 8D*). Broad expression of a rescue gene, or even a fluorescent reporter, often results in subtle changes in locomotion and quiescence that our assays are able to detect. Nevertheless, the data raised the possibility of a ceiling effect for quiescence in these experiments. Two observations suggest that, plausibly, this is not the case: (i) similar differences in undisturbed baseline quiescence were observed between the two *npr-1* mutant alleles, yet both strains exhibited compensation for strong stimuli (*Figure 5E*); and (ii) undisturbed baseline quiescence in all *daf-16* strains was similar to *npr-1(ky13)* mutants and lower than wild-type. Therefore, we favor the interpretation that the function of DAF-16 was required in neurons in our assays. Importantly, the opposing phenotypes of *npr-1* and *daf-16* mutants show that micro-homeostasis, routinely used to stabilize bout architecture in weakly noisy environments, is genetically distinct from the homeostatic responses that strong and stressful disturbances invoke.

## Discussion

Using tunable photo- and mechano-stimulation, we have identified two distinct categories of disruptions to *C. elegans* sleep and characterized the corresponding responses. We have shown that during

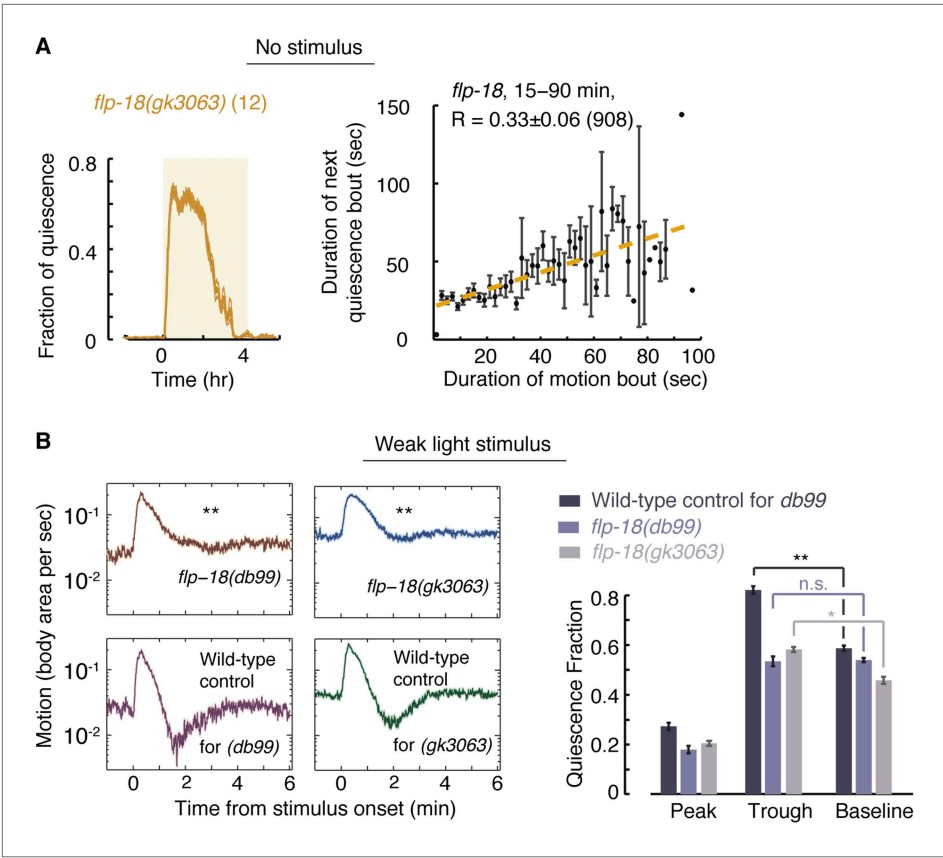

**Figure 7**. FLP-18 plays a role in modulating bout durations in the presence of weak disturbances. (**A**) Posture analysis of undisturbed *flp-18(gk3063)* mutants revealed wild-type-like overall quiescence but reduced correlations between subsequent bouts. R = 0.33 ± 0.06, N = 12 animals. These correlations were significantly different (p < 0.05) from those of wild-type and *npr-1* mutants shown in ***Figures 2A and 5A***, respectively. (**B**) Frame subtraction analysis of *flp-18* mutants during L4leth in the presence of weak blue light stimuli (15 s, 20 mW/cm²). All stimuli were initiated at t = 0. The dynamics of locomotion revealed defects in the ability of *flp-18* mutants to compensate for the motion induced by the stimulus with enhanced quiescence. Left: the locomotion responses during lethargus of each of the two alleles tested and its wild-type control group shown on a semi-log scale. Shaded area denotes mean ± s.e.m. Asterisks denote that during the trough in locomotion, the fraction of quiescence of the mutant allele was significantly lower than that of its respective wild-type control (p < 0.01). Right: for each strain, the quiescence fraction was calculated during 1 min intervals centered at the times of the peak and trough of the L4leth responses, as well as for their respective pre-stimulus baselines. Plots and bars depict mean ± s.e.m obtained from datasets of N = 40–50 animals per condition. Asterisks and double asterisks denote p < 0.05 and p < 0.01, respectively.

The following figure supplements are available for figure 7:

**Figure supplement 1**. A fluorescent reporter of FLP-18 in VC motor neurons and head neurons.

**Figure supplement 2**. Bout correlations in undisturbed flp-18 mutants.

lethargus, motion plays a causal role in modulating the duration of subsequent quiescence. Under low noise conditions, micro-homeostasis manifested as a dependence of the duration of quiescence bouts on the duration and nature of recently preceding motion. The dynamic extension of quiescence bouts depended on the function of an NPY receptor-like protein (NPR-1). However, this did not require DAF-16/FOXO, perhaps because transcriptional level control is typically too slow to respond dynamically on timescales of 10s of seconds (***Yosef and Regev, 2011***). Since biological mechanisms naturally function in a continuous range of conditions, it was both expected and observed that similar mechanisms regulated the compensatory responses in the presence of weak or no stimuli. However, homeostasis in the

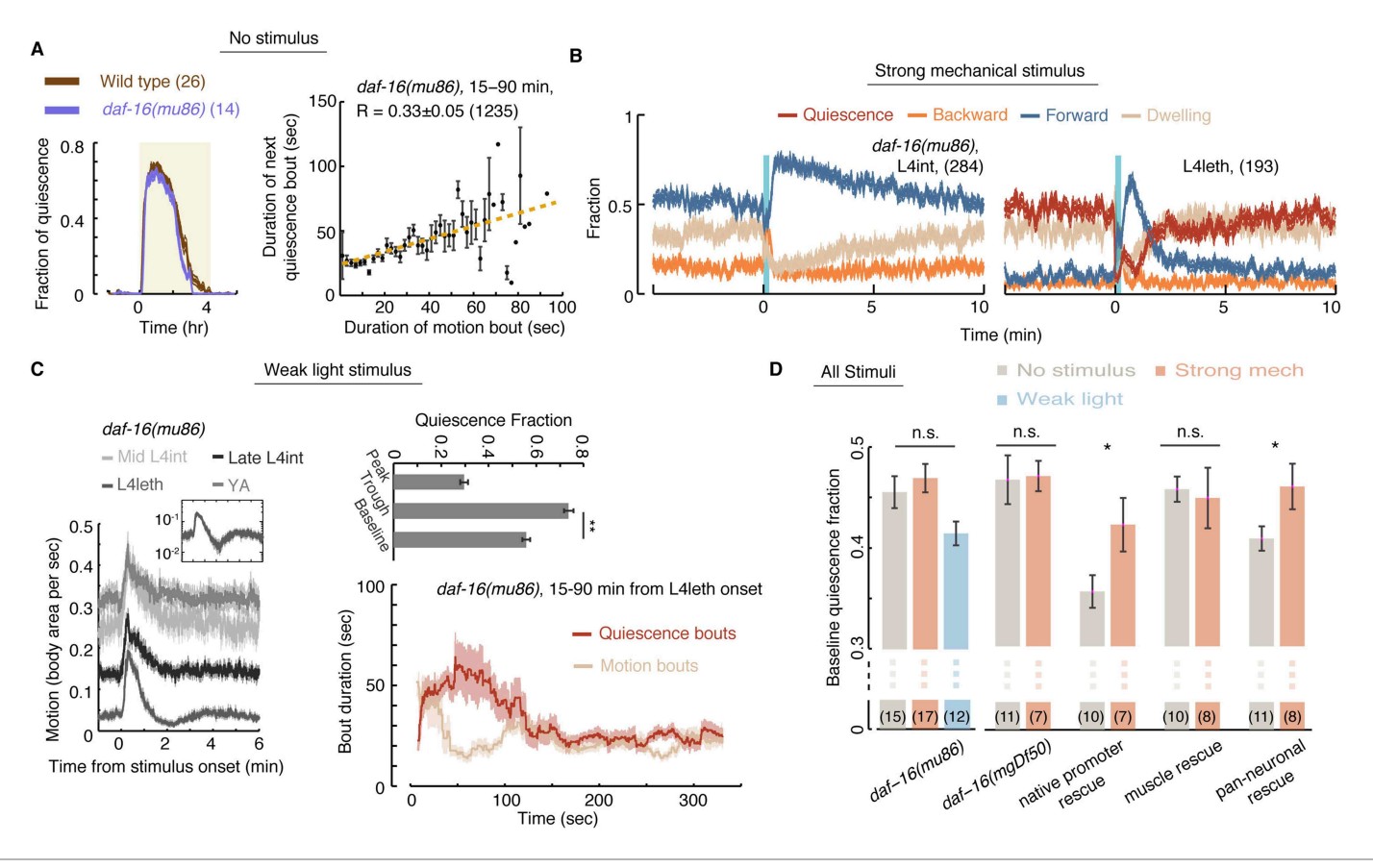

**Figure 8**. Homeostatic responses to strong stimuli, but not micro-homeostasis, require DAF-16. (**A**) Left: the fraction of quiescence of wild-type animals and *daf-16* mutants during L4leth (shaded area). Plots depict mean ± s.e.m, the numbers of animals assayed are denoted in parentheses. Right: pairwise bout correlations shown with a plot of binned bouts (see *Figure 2A* for details). Pairwise correlations were reduced in the mutant, although less so than in *npr-1* mutants (p < 0.05). All correlations are given with 95% confidence intervals and error bars depict ±s.e.m. The number of bouts in each case is denoted in parentheses. (**B**) A posture-based analysis of responses of L4int and L4leth *daf-16* mutants to strong stimuli (15 s, 1 kHz vibrations): the fraction of forward locomotion, backward locomotion, dwelling, and quiescence before, during, and after the stimulus. (**C**) Left: frame subtraction based analyses of responses of L4leth *daf-16* mutants to weak stimuli (15 s, 20 mW/cm², blue light). Inset: the response of *daf-16* mutants during L4leth on a semi-log scale. Middle: the fraction of quiescence during 1 min intervals centered at the times of the peak and trough of the L4leth responses, as well as for their respective pre-stimulus baselines. All stimuli were initiated at t = 0. N = 50–60 animals. Plots and bars depict mean ± s.e.m, asterisks denote p < 0.001. Right: a posture-based analysis of bout dynamics of *daf-16* mutants following a weak stimulus. Plots depict mean ± s.e.m, smoothed using a 30 s running window average. N = 12 animals. The compensatory enhancement of quiescence bouts shortly after the stimulus, as assayed by both methods, was similar to wild-type. (**D**) The mean baseline fractions of quiescence of *daf-16* mutants in undisturbed animals and in the presence of weak and strong stimuli. In contrast to wild-type, baseline quiescence fraction was indistinguishable between the different conditions. Expression of *daf-16* in neurons, but not in body-wall muscles, restored the homeostatic response of *daf-16* mutants to strong mechanical stimuli. Error bar depicts ±s.e.m. The number of stimuli assayed is noted in parentheses for each condition.

presence of strong stimuli was behaviorally and mechanistically distinct. A strong disturbance resulted in the temporary disruption of normal bout dynamics followed by a compensatory upshift of baseline levels of quiescence. These responses did require the function of DAF-16/FOXO but not of NPR-1.

Neuropeptides have been proposed to regulate quiescence during lethargus, but their roles were not examined in detail (*Van Buskirk and Sternberg, 2007*; *Van Buskirk and Sternberg, 2010*; *Nelson et al., 2013*; *Turek et al., 2013*). The apparent discrepancy between the quiescence phenotypes of *egl-3*/*egl-21* and *unc-31* mutants resembles their seemingly contradicting roles in homeostatically ameliorating convulsions caused by cholinergic overexcitation and can be similarly rationalized (*Stawicki et al., 2013*). The mature neuropeptides processed by EGL-3 and EGL-21 are but a subset of the components of dense core vesicles, such that excitatory and inhibitory neuropeptides could act in a combinatorial manner to affect quiescence.

Resuming sleep after a strong or a mild disruption are both common experiences. Subjectively, the two are easily distinguishable, and in both cases the resulting changes to the architecture of sleep reflect homeostatic regulation. Broadly, homeostatic control ensuring adequate sleep amount and quality is a key criterion for sleep-like states (*Tobler, 1983*; *Campbell and Tobler, 1984*; *Sehgal and Mignot, 2011*; *Nelson and Raizen, 2013*; *Tononi and Cirelli, 2014*). Homeostasis in mammalian sleep can be readily observed under disturbed or undisturbed conditions. For instance, the spectral power density associated with slow wave sleep (in the 0.75–4.0 Hz range) decays exponentially during an undisturbed sleep period, while extending the duration of wakefulness enhances it (*Franken et al., 1991*; *Kecklund and Åkerstedt, 1992*). Nevertheless, the sleep literature generally regards sleep homeostasis as a single mechanism (*Borbély, 1982*; *Daan et al., 1984*; *Hendricks et al., 2000*; *Saper et al., 2005*; *Andretic et al., 2008*; *Mackiewicz et al., 2008*; *Cirelli, 2009*; *Crocker and Sehgal, 2010*; *Wang et al., 2011*; *Brown et al., 2012*; *Nelson and Raizen, 2013*; *Porkka-Heiskanen, 2013*). To our knowledge, responses to weak disturbances to sleep were not previously carefully analyzed, and the distinction between routine stabilization and compensation for stressful agitation was not examined in detail.

Despite recent findings in genetically tractable invertebrate models, the understanding of mechanisms that regulate sleep homeostasis remains incomplete (*Andretic et al., 2008*; *Cirelli, 2009*; *Sehgal and Mignot, 2011*; *Driver et al., 2013*; *Shi et al., 2014*). NPY was implicated in the regulation of sleep in humans, rats, fruit flies, and nematodes (*Antonijevic et al., 2000*; *Tóth et al., 2007*; *Dyzma et al., 2010*; *Van den Pol, 2012*; *Choi et al., 2013*; *He et al., 2013*; *Nagy et al., 2014*). In *C. elegans*, the NPY receptor homolog NPR-1 affects a range of responses to external stimuli, as well as innate behaviors such as social feeding and quiescence (*De Bono and Bargmann, 1998*; *De Bono et al., 2002*; *Davies et al., 2004*; *Chang et al., 2006*; *Macosko et al., 2009*; *McGrath et al., 2009*; *Choi et al., 2013*; *Nagy et al., 2014*). Interestingly, NPR-1 was found to play a major role in both lethargus micro-homeostasis (this study) and the homeostatic response to a motoneuron imbalance. In the latter case, NPR-1 was required to compensate for cholinergic over-excitation and GABAergic inhibition that were caused by a gain-of-function in a neuronal nicotinic acetylcholine receptor (*Stawicki et al., 2013*). We hypothesize that these two types of homeostatic responses are closely linked, and further studies will be required to conclusively determine if this is the case.

Recent years have seen a rise in the appreciation of the importance and abundance of peptidergic modulation of neuronal function (*Li and Kim, 2008*; *Bargmann, 2012*; *Marder, 2012*; *Taghert and Nitabach, 2012*; *Holden-Dye and Walker, 2013*). In *C. elegans*, peptidergic regulation was shown to affect quiescence during lethargus (*Nelson et al., 2013*; *Turek et al., 2013*). The apparent discrepancy between the phenotypes of *egl-3*/*egl-21* and *unc-31* mutants suggests that quiescence may be regulated by the combinatorial action of excitatory and inhibitory neuropeptides and that this combinatorial regulation promotes responsive bout dynamics. Our findings are consistent with a model in which activity during lethargus generates a 'pressure' which is ameliorated during periods of quiescence. A particular balance of inhibitory and excitatory neuropeptides may be required for keeping a record of and/or for the process of alleviating this pressure.

Finally, responses to external stimuli during lethargus were different from responses during the L4int and YA stages. In contrast, responses were similar whether the onset of the stimulus coincided with quiescence or motion during lethargus. This suggests that bouts of motion are not analogous to brief intervals of wakefulness. Rather, *C. elegans* sleep may progress through two alternating micro-states.

## Materials and methods

### Strains

*C. elegans* strains were maintained and grown according to standard protocols (*Brenner, 1974*). The following strains were used: wild-type strain N2, Hawaiian CB4856, CB169 *unc-31(e169)*, CB928 *unc-31(e928)*, CX4148 *npr-1(ky13)*, DA609 *npr-1(ad609)*, MT1541 *egl-3(n729)*, MT1241 *egl-21(n611)*, CF1038 *daf-16(mu86)*, GR1307 *daf-16(mgDf50)*, NQ440 *daf-16(mgDf50)*; qnIs42[*Punc-119::GFP::daf-16*; *Pmyo-2::mCherry*], NQ441 *daf-16(mgDf50)*; qnIs45[*Pdaf-16::GFP::daf-16*; *Pmyo-2::mCherry*], NQ145 *daf-16(mgDf50)*; qnEx38[*Pmyo-3::GFP::daf-16*; *Pmyo-2:mCherry*], VC2016 *flp-18(gk3063)*, AX1410 *flp-18(db99)*, AX1444 dbIs[*Pflp-18::flp-18::sl2::gfp*].

## Behavioral assays

Motion and quiescence were identified using previously described methods (*Nagy et al., 2014*). Briefly, animals were grown at 20°C on standard NGM plates seeded with *Escherichi coli* OP50 bacteria. Mid to late L4 individuals were sealed into individual 'artificial dirt' chambers filled with an overnight OP50 culture concentrated 10-fold and resuspended in NGM medium (*Singh et al., 2011*). Animals were imaged at 2 frames per second at a 1.2× magnification for frame subtraction experiments or 10 frames per second at a 4.2× magnification for posture-based analysis using a CCD camera (Prosilica GC2450, Allied Vision Technologies, Stadtroda, Germany). Motion and quiescence were determined as previously described (*Iwanir et al., 2013*; *Nagy et al., 2014*). Frame subtraction data were obtained from the raw images using custom Matlab script (Mathworks Inc., Natick MA) and quiescence was scored when no pixel changed its greyscale value beyond a threshold value (*Husson et al., 2007*) between consecutive frames.

## Posture based behavioral analysis

The precise analysis of animal behavior, based on the identification of the body posture, required high spatial and temporal resolution data. Image analysis and secondary data analysis were performed as previously described using a custom suite of machine vision tools, called PyCelegans, and custom Matlab scripts, respectively (*Nagy et al., 2013*, *2014*). In brief, we identified the body midline in each frame, as well as the positions of the head and the tail. Each midline was divided into 20 equal intervals and the dynamics of the angles between these intervals were used to identify quiescence and directed locomotion states. The onset of lethargus was identified by visual inspection of quiescence data. We note that typical *C. elegans* behavioral assays provide a throughput of 100–1000 animals per day. In contrast, the detailed and computationally intensive posture-based analysis produced a detailed and an accurate account of behavior over 10 hr at a throughput of 3–5 animals per day.

## External stimuli

Blue light (λ = 475 ± 15 nm) was supplied by a Luxeon Star 7-LED assembly with a diffused optic array driven by a 700 mA FlexBlock driver. The LED assembly was mounted to the scopes approximately 7 cm from the sample location. Light intensity was measured at the location of the animals. The timing of light stimuli was controlled using LabView (National Instruments Inc., Austin TX). Mechanical stimuli were generated using 50 mm piezo buzzer elements (Digikey part no. 668-1190-ND) as previously described (*Nagy et al., 2014*). The timing and duration of the stimuli were controlled using a custom Matlab script. An external stimulus was provided every 15 min throughout the course of each experiment. Animals resumed baseline behavior dynamics after no more than 5 min after each individual stimuli and no habituation was observed in the responses to the repeated stimuli.

## Fluorescent expression reporter

The *Pflp-18::flp-18::SL2::gfp* reporter strain was a kind gift from the de Bono lab (*Cohen et al., 2009*). Late L4int larvae were placed in an artificial dirt microfluidic device with an overnight OP50 culture concentrated 10-fold and resuspended in NGM medium (*Lockery et al., 2008*). Epi-fluorescence images of the freely behaving animals were acquired for 5 s every 15 min, for 8–9 hr, at a magnification of 20× and a frame rate of 4 frames per second. Regions of interest containing the neurons were identified by visual inspection. Fluorescence was quantified as the sum of pixel intensities that were higher than one standard deviation above the mean of the background pixel intensity. The background was calculated from a region of the body of the animal that was proximal to the neuron of interest but did not contain it. Under these conditions, no photo-bleaching was detected.

## Statistical and numerical analysis

Data analysis was performed using custom Matlab (Mathworks Inc., Natick, MA) scripts. For comparisons in summary statistics panels, significance was calculated using a one-way ANOVA test. Post-hoc correction for multiple comparisons was performed using the Bonferroni adjustment. Correlation coefficients are presented with 95% confidence intervals (Matlab statistical toolbox). Corresponding p-values are the probabilities of obtaining the observed correlation by chance, when the true correlation is zero. To graphically demonstrate pairwise correlations between durations of bouts in *Figures 1E, 4B, 6A, and 7C*, we grouped all bouts of motion in order of ascending duration in bins of 2 s and used a linear fit as a guide for the eye. The correlation coefficients were calculated using the original pairs of bout durations (as opposed to the binned data). Bout correlations of wild-type

animals, *flp-18* mutants, and *npr-1* mutants during the period 15–90 min from the onset of L4leth were compared by applying Fisher's z-transformation and calculating the 95% confidence interval for the difference of the correlation coefficients as described in *Zou (2007)*.

## Acknowledgements

Some strains were provided by the CGC, which is funded by NIH Office of Research Infrastructure Programs (P40 OD010440). This work was supported by the Burroughs Wellcome Fund Career Award at the Scientific Interface (DB), the Searle Scholars Program (DB), NSF GRFP (DGE-0638477, NT), and the NSF (IOS 1256989). We thank D Raizen for strains and for useful discussions, and E Efrati and L Avery for useful discussions.

## Additional information

### Funding

| Funder | Grant reference number | Author |
|---|---|---|
| National Science Foundation | IOS 1256989 | David Biron |
| Burroughs Wellcome Fund | CASI award | David Biron |
| Kinship Foundation | Searle Scholars Program | David Biron |
| National Science Foundation | DGE-0638477 | Nora Tramm |

The funders had no role in study design, data collection and interpretation, or the decision to submit the work for publication.

### Author contributions

SN, NT, JS, Conception and design, Acquisition of data, Analysis and interpretation of data, Drafting or revising the article; SI, EL, Conception and design, Drafting or revising the article; IAS, Conception and design, Acquisition of data, Analysis and interpretation of data; DB, Conception and design, Analysis and interpretation of data, Drafting or revising the article

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
