## [Decision Letter]

Thank you for sending your work entitled “Homeostasis in *C. elegans* sleep is characterized by two behaviorally and genetically distinct mechanisms” for consideration at *eLife*. Your article has been favorably evaluated by Eve Marder (Senior editor) and 3 reviewers, one of whom, Ronald L Calabrese, is a member of our Board of Reviewing Editors.

The Reviewing editor and the other reviewers discussed their comments before we reached this decision, and the Reviewing editor has assembled the following comments to help you prepare a revised submission.

The authors present a behavioral analysis of homeostasis during the sleep-like state of lethargus in *C. elegans*. They find two distinct levels of homeostasis separable phenomenologically by the response to disturbing stimuli. Micro-homeostasis occurs in undisturbed or weakly disturbed (stimulated) animals and homeostasis occurs in strongly disturbed (stimulated) animals. The behavioral data set is extensive and thorough and for the main part the data are appropriately interpreted. They then begin to differentiate between these two forms of homeostasis genetically. While some of these genetic data are essential to differentiate the responses there needs to be a rescue experiment to increase certainty of the specificity of the *flp-18* mutant response and some of the other genetic data should be deleted as too preliminary (expression experiment for *flp-18* and *unc-31*, *dgk-1*, and *goa-1* mutant analyses). The authors should be aware that *eLife* has Advances, so that an additional part of the story that takes it to the next step (e.g., a more in depth mutant analysis) could be linked to this paper, if it is accepted by *eLife*.

In revision the authors should focus on the following:

1) The three reviewers agree that the observation in *C. elegans* of two homeostatic mechanisms in sleep disturbance is a significant advance as homeostasis is generally regarded as one mechanism in the mammalian sleep literature. This should be shored up by attention more generally to the sleep literature in the Discussion.

2) The paper is not written for a general audience of educated biologists. It is filled with undefined acronyms and abbreviations and with the jargon of *C. elegans*. Make sure to define all terms, e.g. YA, L4int etc. The Results section is especially overly terse; please explain the sophisticated behavioral analyses more thoroughly, adding additional explanatory figures if necessary.

3) The reviewers agree that the authors are far from establishing mechanisms for either type of homeostatic response. What is important about the genetic experiments is their use to show that the two responses are controlled differently: this is a novel notion with relevance to other animals. The analysis aimed at mechanisms (e.g. *flp-18* reporter expression, g-protein pathway analysis) is generally weak and a distraction from the above punch line. Some aspects of *flp-18* genetic should remain, i.e. mutant analysis, but this needs to be shored up with rescue experiments. The *flp-18* strain used, VC2016 is a heavily mutagenized strain carrying multiple mutations. There is a strain (AX1444) carrying a wild-type *flp-18* transgene, which you could use for a rescue experiment. The *flp-18* expression analysis is weak and should be removed. The *unc-31*, *dgk-1*, and *goa-1* analyses are incomplete and should be deleted.

4) We are also concerned that the conclusions go too far based on the current genetic analyses. The data make it safe to conclude that genetic perturbations support the novel hypothesis that there are two homeostatic processes regulating responses to sleep disturbance. It is not warranted yet to argue that *npr-1* regulates one of these processes.

5) One thing that was very confusing and needs clarification is the data of Figure 3. Why was baseline quiescence measured before the stimulus? Is the stimulus repeated and there is a cumulative effect? How is this a meaningful measure? Shouldn't the post-stimulus baseline be compared to the pre-stimulus baseline? Other figures use the same measure. This must be explained and justified.

[Editors' note: further revisions were requested prior to acceptance, as described below.]

Thank you for sending your work entitled “Homeostasis in *C. elegans* sleep is characterized by two behaviorally and genetically distinct mechanisms” for consideration at *eLife*. Your article has been favorably evaluated by Eve Marder (Senior editor), Ronald L Calabrese (Reviewing editor), and 2 reviewers.

The Reviewing editor and the reviewers discussed their comments before we reached this decision, and the Reviewing editor has assembled the following comments to help you prepare a revised submission.

The authors present a behavioral analysis of homeostasis during the sleep-like state of lethargus in *C. elegans*. They find two distinct levels of homeostasis separable phenomenologically by the response to disturbing stimuli. Micro-homeostasis occurs in undisturbed or weakly disturbed (stimulated) animals and homeostasis occurs in strongly disturbed (stimulated) animals. The behavioral data set is extensive and thorough and for the main part the data is appropriately interpreted.

The previous review addressed several concerns, which the authors have now addressed. One concern was over the presentation and its accessibility to a general reader of *eLife*. This concern has been substantially addressed so that the manuscript is now more readable for biologists outside the *C. elegans* community. Another concern about the context of the results in the field of sleep research has been addressed by additions to Discussion. Concerns about *flp-18* genetics and expression have also been addressed.

Concerns about the behavioral analyses were addressed by more clearly and explicitly stating what was done but there are still some important clarifications to be made. There is clear consensus on this issue. To quote one of the expert reviewers: “The schematic at the top of Figure 3 is helpful in understanding the authors' stimulation and analysis protocol. However, nowhere in the figures or supplementary figures could I find a clear example showing the elevated baseline quiescence in the 5 minutes prior to the next stimulus. Figures 4 and 8 shows the 5 minutes prior to the strong mechanical stimulation, and then the 10 minutes after the stimulus. The 5 minutes of baseline quiescence before the stimulus looks to my eye the same as the 10 minutes of quiescence after the stimulus. Isn't the relevant period 10-15 minutes *after* the stimulus? This is the new baseline prior to the next stimulus. Also, if the strong stimulus is repeated every 15 minutes, and there is a consequent elevation of 'baseline quiescence' in the 5 minutes prior to the next stimulation, is the subsequent 'baseline quiescence' further elevated? That is, is the effect of stimulations cumulative? Showing the data in 4B and 8B with the longer time scale shown in 3A may be helpful.”

There are serious concerns by all reviewers on some sentences in which claims are made about the role of NPR-1 in homeostasis. “Thus, in addition to the phenotypic differences described above, homeostatic compensation during undisturbed or weakly disturbed lethargus was regulated by NPR-1, while homeostatic compensation for strong stimuli was not. Thus, the routine stabilization of lethargus behavior in low-noise environments and the homeostatic compensation for stressful disturbances are mechanistically separable. These findings are consistent with a model in which NPR-1 dynamically modulates quiescence during lethargus in response to spontaneous or induced mild variations in locomotion.” Specifically the phrase '...regulated by NPR-1...' must be altered to something more in line with the data like '...affected by...' or '...required...' and the word 'dynamically' removed ('...dynamically modulates...').

The section on Combinatorial Peptide Action in the Results section overstates the case and needs to be cut back. You have observed inconsistent effects of different general peptide secretion mutants and conclude that it is combinatorial. The authors should simply report their observations in Results without the current heading and potentially include one sentence in the Discussion saying that the different phenotypes of *egl* and *unc* mutations can be rationalized by postulating multiple peptides with positive and negative effects.

Moreover in the opinion of the one reviewer the genetic analysis remains superficial and the paper would benefit from further restricting claims made on the basis of these experiments

These serious concerns must be addressed; this should require only minor re-writing and can be handled in the future by the BRE.

---

## [Author Response]

*1) The three reviewers agree that the observation in C. elegans of two homeostatic mechanisms in sleep disturbance is a significant advance as homeostasis is generally regarded as one mechanism in the mammalian sleep literature. This should be shored up by attention more generally to the sleep literature in the Discussion*.

A paragraph was added to the Discussion section in order to explicitly point out that homeostasis is generally regarded as one mechanism in the mammalian sleep literature. Relevant articles from the sleep literature, ranging from the introduction of the canonical model homeostatic process to recent reviews, were cited.

*2) The paper is not written for a general audience of educated biologists. It is filled with undefined acronyms and abbreviations and with the jargon of* C. elegans*. Make sure to define all terms, e.g. YA, L4int etc.*

The Introduction was revised such that it now includes definitions of abbreviations and *C. elegans* jargon.

*The Results section is especially overly terse; please explain the sophisticated behavioral analyses more thoroughly, adding additional explanatory figures if necessary*.

An explanatory panel was added to Figure 3, containing a diagram that outlines the design of the repeated stimulus assay used throughout this study. A corresponding paragraph was added to the Results section. Specifically, the diagram and the text explain that the 5 minute period that preceded each stimulus had also started 10 minutes after the preceding stimulus.

In addition, the original Figure 3 was divided to two figures: the new Figure 3 depicting photostimulation data and the new Figure 4 depicting (mainly) mechanostimulation data. The text was revised accordingly.

*3) The reviewers agree that the authors are far from establishing mechanisms for either type of homeostatic response. What is important about the genetic experiments is their use to show that the two responses are controlled differently: this is a novel notion with relevance to other animals. The analysis aimed at mechanisms (e.g.* flp-18 *reporter expression, g-protein pathway analysis) is generally weak and a distraction from the above punch line. Some aspects of* flp-18 *genetic should remain, i.e. mutant analysis, but this needs to be shored up with rescue experiments.*

The *Pflp‐18* promoter drives expression in approximately 20 head neurons and in cholinergic motor neurons. Expression of *flp‐18* was reported to be generally lower in the motor neurons. The AX1444 reporter strain exhibits defects in locomotion and quiescence, plausibly due to overexpression of *flp‐18* in an unidentified subset of these neurons. Such defects would confound the interpretation of the simple rescue experiment suggested by the reviewers.

However, we obtained mutants carrying a second allele, *flp‐18(db99)*, and assayed them using the frame subtraction method. This 2kb deletion of a region spanning the promoter and the first two exons (including the initiation codon) is likely to be a strong loss‐of‐function or null allele [[17], Cell Met]. In this assay, compensation deficiency of *flp‐18(db99)* mutants was stronger than that of *flp‐18(gk3063)* mutants and comparable to that of *npr‐1* mutants. These results validated the role of *flp‐18* peptides in the homeostatic response for weak stimuli, although they came short of conclusively excluding roles for additional peptides.

*The* flp-18 *strain used, VC2016 is a heavily mutagenized strain carrying multiple mutations. There is a strain (AX1444) carrying a wild-type* flp-18 *transgene, which you could use for a rescue experiment. The* flp-18 *expression analysis is weak and should be removed.*

The analysis of *flp‐18* expression, which was based on a fluorescent reporter, can and will be removed from the manuscript at the discretion of the reviewers. However, we feel that taken together with the mutant analysis these results strengthen the case for a lethargus‐specific role for *flp‐18* peptides in homeostatic responses. Therefore, the current version includes these results as supplementary figures accompanied by two paragraphs in the text. Before the final decision to remove these results is reached, we would like to offer the following considerations:

In *C. elegans*, GFP has a fast maturation rate of about ½ hr (see, e.g., Boulin, Etchberger, and Hobert, Wormbook, 2006), such that the kinetics of maturation are not expected to strongly affect our results. In addition, the complete absence of change in fluorescence in head neurons provides an excellent internal control for this measurement. Importantly, since the change in expression is expected to occur only in the motor neurons a subset of *flp‐18* expressing neurons; changes in expression are not expected to be detectable above noise using qPCR analysis of whole animals. In agreement with these considerations, a recently published profiling of the temporal expression pattern of *flp‐18* in *C. elegans* used the very same reporter strain and did not perform qPCR [Stawicki et al., PLoS Gen, 2013]. Our data supports the idea that these phenomena may be linked.

*The* unc-31*,* dgk-1*, and* goa-1 *analyses are incomplete and should be deleted.*

The analysis of *unc‐43*, *dgk‐1,* and *goa‐1* was removed as suggested by the reviewers.

However, NPR‐1 is an extremely well-studied neuropeptide receptor. Therefore, the *unc-31* results implicating neuropeptides in the flexibility/responsiveness of bout dynamics are highly relevant to the manuscript. Since two independent alleles of *unc‐31* were assayed, the confidence that the observed phenotype was caused by the *unc‐31* mutation is (at least) comparable to a rescue experiment with an innate promoter – see response above. In the revised version, the discussion of the *unc‐31* data was moved to the Results section and its relevance to the NPR‐1 results was explicitly noted.

These results suggest an apparent discrepancy when compared to the *egl‐3*/*egl‐21* phenotypes, both in our hands and as previously reported (83). We felt that this issue could and should be addressed. Therefore, it is discussed in the Results section. We note that in this case the identical phenotypes of *egl‐3* and *egl‐21* served the same purpose as assaying two independent alleles.

*4) We are also concerned that the conclusions go too far based on the current genetic analyses. The data make it safe to conclude that genetic perturbations support the novel hypothesis that there are two homeostatic processes regulating responses to sleep disturbance. It is not warranted yet to argue that npr-1 regulates one of these processes*.

Assaying two independent mutant alleles of the same gene is considered to be at least as reliable as a general rescue experiment under a native promoter, since rescues can be partial and subject to various caveats. It is standard in the field to use the two approaches interchangeably (as reflected by one of the specific comments: "testing a second allele…would be required to prove that the phenotype is explained by the mutation in the gene of interest.”).

As noted by reviewer #3, we originally presented an analysis of a single *daf‐16* allele and erroneously referred to related published data obtained from a second allele (Driver el al., 2013). David Raizen has kindly provided us with transgenic strains that were used in his study and we assayed the second *daf‐16* allele, a native promoter rescue strain, a pan‐neuronal rescue strain, and a body‐wall muscle rescue strain. In brief, our data suggest that the function of DAF‐16 in neurons, but not in muscles, is required for homeostatic compensation for strong disturbances. These results are described in the Results section and in Figure 7 of the revised manuscript.

*5) One thing that was very confusing and needs clarification is the data of*
Figure 3*. Why was baseline quiescence measured before the stimulus? Is the stimulus repeated and there is a cumulative effect? How is this a meaningful measure? Shouldn't the post-stimulus baseline be compared to the pre-stimulus baseline? Other figures use the same measure. This must be explained and justified*.

The stimulus was repeated (every 15 minutes), such that the baseline was calculated 10‐15 minutes *after* the stimulus. An explanatory panel was added to Figure 3 and the manuscript was revised accordingly.

[Editors' note: further revisions were requested prior to acceptance, as described below.]

*Concerns about the behavioral analyses were addressed by more clearly and explicitly stating what was done but there are still some important clarifications to be made. There is clear consensus on this issue. To quote one of the expert reviewers “The schematic at the top of*
Figure 3
*is helpful in understanding the authors' stimulation and analysis protocol. However, nowhere in the figures or supplementary figures could I find a clear example showing the elevated baseline quiescence in the 5 minutes prior to the next stimulus.*
Figures 4 and 8
*shows the 5 minutes prior to the strong mechanical stimulation, and then the 10 minutes after the stimulus. The 5 minutes of baseline quiescence before the stimulus looks to my eye the same as the 10 minutes of quiescence after the stimulus. Isn't the relevant period 10-15 minutes* after *the stimulus? This is the new baseline prior to the next stimulus. Also, if the strong stimulus is repeated every 15 minutes, and there is a consequent elevation of 'baseline quiescence' in the 5 minutes prior to the next stimulation, is the subsequent 'baseline quiescence' further elevated? That is, is the effect of stimulations cumulative? Showing the data in 4B and 8B with the longer time scale shown in 3A may be helpful.”*

The stimulus was repeated every 15 minutes and the resulting behavioral dynamics were averaged over many such cycles. Therefore, in the data presented the distinction between “5 minutes before” and “10 minutes after” a stimulus is not a meaningful one. In the averaged data, both “5 minutes before…” or “10 minutes after…” would refer to the very same time point, namely the time point at 2/3 of the interval between two consecutive stimuli. In other words, the averaged data has periodic boundaries. Graphically, this can be represented by plotting this data on a circular time axis where “before” and “after” are indistinguishable (as long as the fraction of quiescence is in its steady state).

During lethargus, the short-term response to each stimulus lasted 3 minutes. After this short-term response was complete, the fraction of quiescence returned to a steady state characteristic of the conditions of the experiment. The fraction of time spent in quiescence during this steady state was defined as the baseline level of quiescence for the relevant experimental conditions. In this framework, baselines were compared between different conditions. Elevation of the steady state quiescence, as compared to the mean quiescence of undisturbed animals, constituted the sustained response to the presence of repeated strong stimuli.

A supplementary figure was added to Figure 3 and the text was revised to clarify these points.

*There are serious concerns by all reviewers on some sentences in which claims are made about the role of NPR-1 in homeostasis. “Thus, in addition to the phenotypic differences described above, homeostatic compensation during undisturbed or weakly disturbed lethargus was regulated by NPR-1, while homeostatic compensation for strong stimuli was not. Thus, the routine stabilization of lethargus behavior in low-noise environments and the homeostatic compensation for stressful disturbances are mechanistically separable. These findings are consistent with a model in which NPR-1 dynamically modulates quiescence during lethargus in response to spontaneous or induced mild variations in locomotion.” Specifically the phrase '...regulated by NPR-1...' must be altered to something more in line with the data like '...affected by...' or '...required...' and the word 'dynamically' removed ('...dynamically modulates...')*.

The required changes were made to the text and the Abstract.

*The section on Combinatorial Peptide Action in the Results section overstates the case and needs to be cut back. You have observed inconsistent effects of different general peptide secretion mutants and conclude that it is combinatorial. The authors should simply report their observations in Results without the current heading and potentially include one sentence in the Discussion saying that the different phenotypes of* egl *and* unc *mutations can be rationalized by postulating multiple peptides with positive and negative effects*.

The required changes were made to the text.